# Field Evaluation of Commercial Vaccines against Infectious Bovine Rhinotracheitis (Ibr) Virus Using Different Immunization Protocols

**DOI:** 10.3390/vaccines9040408

**Published:** 2021-04-20

**Authors:** Laureana De Brun, Mauro Leites, Agustín Furtado, Fabricio Campos, Paulo Roehe, Rodrigo Puentes

**Affiliations:** 1Departamento de Patobiología, Facultad de Veterinaria, Universidad de la República, Montevideo 11600, Uruguay; laureana@fvet.edu.uy (L.D.B.); maurolei@gmail.com (M.L.); afurtado@fvet.edu.uy (A.F.); 2Laboratório de Bioinformática & Biotecnologia, Campus de Gurupi, Universidade Federal do Tocantins, Gurupi 77410-530, Brazil; camposvet@gmail.com; 3Instituto de Ciências Básicas da Saúde, Universidade Federal do Rio Grande do Sul, Rio Grande do Sul, Porto Alegre 90050-170, Brazil; proehe@gmail.com

**Keywords:** BoHV-1, BoHV-1 vaccines in a field herd, Clostridial, FMD

## Abstract

*Bovine alphaherpesvirus 1* is ubiquitous in cattle populations and is associated with several clinical syndromes, including respiratory disease, genital disease, infertility and abortions. Control of the virus in many parts of the world is achieved primarily through vaccination with either inactivated or live modified viral vaccines. The objective of this study was to evaluate the performance of four commercially available BoHV-1 vaccines commonly used in Central and South America. Animals were divided into eight groups and vaccinated on days 0 and 30. Groups 1 to 4 received two doses of four different BoHV-1 commercial vaccines (named A to D). Groups 5 and 6 received vaccine D plus a vaccine for either Clostridial or Food-and-Mouth-Disease (FMD), respectively. Group 7 received one dose of two different brands of reproductive vaccines. Serum samples were collected from all animals on days 0, 30 and 60 to evaluate neutralizing and isotype-specific (IgG1 and IgG2) antibodies. Of the four commercial vaccines evaluated, only vaccine A induced neutralizing antibodies to titers ≥ 1:8 in 13/15 (86%) of the animals 60 days post-vaccination. Levels of IgG2 antibody increased in all groups, except for group 2 after the first dose of vaccine B. These results show that only vaccine A induced significant and detectable levels of BoHV-1-neutralizing antibodies. The combination of vaccine D with Clostridial or FMD vaccines did not affect neutralizing antibody responses to BoHV-1. The antibody responses of three of the four commercial vaccines analyzed here were lower than admissible by vaccine A. These results may be from vaccination failure, but means to identify the immune signatures predictive of clinical protection against BoHV-1 in cattle should also be considered.

## 1. Introduction

*Bovine alphaherpesvirus 1* (BoHV-1) produces a wide variety of clinical manifestations, including infectious bovine rhinotracheitis (IBR), infectious pustular vulvovaginitis and balanoposthitis (IPV and IPB), infertility, abortion and systemic infection in newborn calves, collectively causing substantive economic losses for cattle production chains worldwide. BoHV-1 isolates are taxonomically divided into three different genotypes and subgenotypes, based on their antigenic and genomic characteristics, named BoHV-1.1, BoHV-1.2a and BoVH-1.2b [1,2].

Following infection, BoHV-1 (like other herpesviruses) establishes latent infections in neuronal ganglia [3,4]. During latency, the virus remains in a non-replicative state in the form of episomes in the nuclei of infected cells. Upon reactivation, infectious virions can then be shed and become a source of infection to other animals. Reactivation can be experimentally induced by several stimuli, such as transport, parturition and administration of glucocorticoids [5]. This peculiarity allows the virus to go undetected and thus persist in herds, ensuring perpetuation of the virus [6].

The infection is widely distributed thought the world, with large differences in prevalence between countries. It is present in nearly all the countries of the Americas, Australia, New Zealand and some countries of Europe, Asia and Africa. In Uruguay, the disease is highly prevalent [7]. When BoHV-1-infected bulls are introduced into a herd, it has a negative impact on reproductive performance from low conception rate to abortions [8]. Vaccination for BoHV-1 is thought to minimize reproductive losses and prevent clinical signs in cattle, such as respiratory symptoms and conjunctivitis. A recently published meta-analysis shows that the BoHV-1 vaccine is associated with a 60% decrease in the risk of bovine abortion [9]. Inactivated BoHV-1 vaccines can be safely used in pregnant animals and so are widely used in breeding herds with other vaccines for pathogens, such as Bovine Viral Diarrhea (BVD), *Leptospira spp* and *Campylobacter fetus*. Other immunogens, such as Clostridial antigens, and Foot and Mouth Disease Virus (FMDV) are often applied simultaneously with vaccines against infectious diseases that affect reproduction in cattle. However, it is not clear whether such immunization protocols interfere with the immune response against BoHV-1.

Primary BoHV-1 infection induces strong humoral and cell-mediated immune responses in cattle, as would be expected under normal conditions [10,11]. The serum neutralization (SN) test used to quantify BoHV-1 neutralizing antibodies in sera of cattle is recommended by the World Organization for Animal Health (OIE). SN has been used extensively to evaluate vaccines [11,12,13,14], although there is a difference in the titers obtained according to the sample used [15]. Moreover, it has been shown that the efficacy of the immune response against BoHV-1 depends on increased Th1-type cell-mediated responses [16]. The measurement of subclass-specific (IgG1 and IgG2) levels in serum allows an indirect estimate of the profile of the cellular immune response induced after vaccination. Thus, elevated IgG1 levels are related to a Th2 response profile, while high IgG2 levels are associated with a Th1 profile [17,18].

The aim of this study was to evaluate the effectiveness of four inactivated BoHV-1 vaccines commonly available in Uruguay by measuring levels of neutralizing antibodies and specific immunoglobulin subclasses induced by the vaccines. Responses were also evaluated when Clostridial or Foot and Mouth Disease (FMD) vaccines were applied simultaneously.

## 2. Materials and Methods

### 2.1. Animals

The experiment was carried out in a livestock farm located in the Lavalleja Department, Uruguay. From a herd with approximately 1000 Herefords, 120 heifers seronegative against BoHV-1 were selected and divided into eight groups of 15 animals each. The experimental protocols for the cattle studies performed in this paper were approved by Animal Use Ethics Commission (CEUA) from the Universidad de la República Oriental del Uruguay (approval number CEUAFVET-972, Exp. 111900-001057-19).

### 2.2. Vaccination Protocols

The animals were divided into 8 groups of 15 animals each. Groups 1 to 4 received two doses (day 0 and day 30) of different commercial BoHV-1 vaccines. Groups 5 and 6 received a BoHV-1 vaccine in combination with a vaccine for Clostridial (double dose—days 0 and 30) or FMD (single dose—day 0) and group 7 received different brands of reproductive vaccine in each dose. The control group (group 8) was not vaccinated and they were kept in the same herd as the vaccinated animals. (Table 1).

Blood was collected by coccygeal venipuncture on days 0, 30 and 60 from all animals and sera were stored at −20 °C until processing.

### 2.3. Vaccines

Six different commercial vaccines with the following antigens were used:Vaccine A: inactivated suspension of BoHV-1, bovine viral diarrhea virus (BVDV)-1, *Campylobacter fetus*, *Histophilus somni* and *Leptospira spp* in aluminum hydroxide adjuvant.Vaccine B: inactivated suspension of BoHV-1, BoHV-5; BVDV-1, BVDV-2; *Leptospira spp* and *Campylobacter fetus* in aluminum hydroxide adjuvant.Vaccine C: inactivated BoHV-1, BVDV-1, *Campylobacter fetus*, *Leptospira spp* and *Histophilus somni* in aluminum hydroxide adjuvant.Vaccine D: inactivated suspension of BoHV-1, BoHV-5, BVDV-1, BVDV-2, *Leptospira spp* and *Campylobacter fetus* in oil—adjuvant.Clostridial vaccine: Bacterin-toxoid combined with a water-soluble adjuvant containing Clostridium chauvoei, Clostridium septicum, Clostridium haemolyticum, Clostridium novyi, Clostridium sordellii, Clostridium perfringens types B and C and Clostridium perfringens type D.FMD vaccine: Oil emulsion vaccine containing inactivated Foot and Mouth Disease Virus types O1 Campos and A24 Cruzeiro.

For informational purposes, the reproductive vaccines used were (listed in alphabetical order): BIOABORTOGEN^®^ H—Biogénesis—Bagó, Buenos Aires, Argentina; BOVISAN TOTAL Se^®^—Virbac, Montevideo, Uruguay; TRANSVAC^®^—Merial, Buenos Aires, Argentina; and VAC-SULES REPRODUCTIVA FORTE^®^—Microsules, Montevideo, Uruguay. The order of appearance of the marks does not necessarily coincide with the description A to D.

### 2.4. Serum Neutralization (SN) Test

Serum samples were submitted to a serum neutralization (SN) test for detection of antibodies to BoHV-1, according to the OIE protocol [19]. Briefly, 50 µL of a two-fold dilution of each serum was mixed with an equal volume of a suspension containing 100 TCID_50_ (50% tissue culture infectious dose) of BoHV-1 Los Angeles strain (LA—BoHV-1.1). After 24 h incubation at 37 °C, 50 µL of a cell suspension (3 × 10^4^ cells) was added to wells and plates incubated at 37 °C in a 5% CO_2_ incubator. Plates were examined for cytopathic effect (CPE) using a light microscope daily for five days. Antibody titers were expressed as the reciprocal of the highest serum dilution that prevented the induction of CPE after 5 days of incubation. SN antibody titers ≥ 1:8 was used as the cut-off for protection [20].

### 2.5. Indirect ELISA

For the quantification of BoHV-1-specific total IgG, IgG1 and IgG2, three indirect ELISAs were developed. A stock ELISA antigen was prepared on MDBK cells infected with BoHV-1 LA strain at a multiplicity of infection (MOI) between 0.1 and 1 and further treated with 0.2% N-octyl-glucopyranoside (OGP) following previously described methods [11]. Class and subclass-specific peroxidase conjugates (anti-bovine IgG, IgG1 and IgG2 antibodies) were purchased commercially and employed following the manufacturers’ instructions (Jackson Immunoresearch Laboratories INC, West Grove, PA, USA). The plates were coated with a 1:1500 dilution of the antigen in bicarbonate buffer overnight at 4 °C. After the adsorption of the antigen, plates were washed once with 100 µL of PBST-20 (0.5% Tween 20 in 83.3 mM KH_2_PO_4_, 66 mM Na_2_HPO_4_, 14.5 mM NaCl), filled with another 100 µL of PBST-20 and allowed to stand at room temperature for 1 h. For testing, sera were diluted 1:2 in PBST-20 and added to wells in duplicate. After 1 h incubation at 37 °C, plates were washed three times with PBST-20. Class or subclass-specific peroxidase conjugates (diluted in PBS as previously titrated) were then added to wells and the plates were again incubated for 1 h at 37 °C. After another series of washings with PBST-20, 100 µL of the substrate ortho-phenylenediamine (OPD; Sigma, Darmstadt, Germany) with 0.03% H_2_O_2_ were added to plates [11]. After 30-min incubation at room temperature, the optical density (OD) was determined at 405 nm in a Multiskan (Titertek, Waltham, Massachusetts, EUA) ELISA reader.

### 2.6. Statistical Analysis

Fisher’s exact test was used in the analysis of contingency tables and a Student’s *t* test was conducted to compare the means between groups for the different isotypes considering the time factor (GraphPad Prism 6). The significance level was *p* < 0.05.

## 3. Results

The results of the neutralizing antibody titrations of each experimental group are shown in Table 2 and a supplemental file is provided with the complete results. Only vaccine A induced antibody titers ≥1:8 in most animals (86%) at day 60.

The indirect ELISA for total antibodies (*p* < 0.05) relative to IgG1 and IgG2 against BoHV-1 demonstrated that commercial vaccine A used in group 1 induced higher titers of antibodies (*p* < 0.05) compared to the groups that received vaccines C and D, while vaccine B did not induce detectable antibodies in any of the samples collected (Figure 1A).

The subclass-specific antibody revealed a low IgG1 response in all groups (Figure 1B), while IgG2 showed a significant increase in groups 1, 3 and 4 on day 60 post-immunization (Figure 1C).

Additionally, the antibody response against BoHV-1 was evaluated when other vaccines were given simultaneously (Clostridial—group 5 or FMD—group 6). At day 30, the animals immunized with BoHV-1 + Clostridial or BoHV-1 + FMD produced a significantly lower antibody response than the group immunized only with BoHV-1. However, on day 60, this difference had disappeared (Figure 2A). For the antibody subclasses analyzed, a weak IgG1 response was evident in all groups at 30 and 60 days after immunization (Figure 2B), while IgG2 showed a significant increase in vaccinated groups 4, 5 and 6 on day 60 of the experiment (Figure 2C).

The antibody titer showed that on day 60, the titers were significantly different when the vaccine brand was changed between the first and second dose (group 7—vaccine D on day 0 and vaccine A on day 30). At day 60, vaccine A induced the highest antibody titers, while vaccine D induced the lowest antibody titers (*p* < 0.05). When the vaccine brand between both immunizations was changed (group 7), the response at day 60, an intermediate response between the response of vaccine A and D (Figure 3A), was achieved. For the antibody subclasses analyzed, IgG1 level was not different between the groups (Figure 3B), while the IgG2 values at day 60 were significantly higher in group 7 (Figure 3C).

## 4. Discussion

Uruguay has one of the highest beef and milk production volumes per capita in the world, as well as being one of the largest consumers of beef per inhabitant. Thus, livestock production is not only of economic importance but also is a major part of the diet of the country’s population. As such, animal health is a substantial concern for the production chain in the management system regularly employed in Uruguay’s cattle farms. At the production level, there is discussion about the efficacy and durability of the response obtained with the available reproductive vaccines, despite the ample number of recognized national and international vaccines available. Here, different protocols have been developed to analyze the antibody response against BoHV-1 using multivalent commercial vaccines in a field herd. It is necessary to clarify that none of the BoHV-1 vaccines tested in this study were monovalent, although the objective here was to evaluate the response of these multivalent vaccines against BoHV-1. In addition, the responses to the Clostridial and FMDV in this study have not been evaluated.

At the international level, vaccines that protect against BoHV-1 are expected to generate at least a neutralizing antibody titer of ≥1:8 [20], although it cannot be ruled out that immunization may still have a protective effect in animals with medium or low antibody titers. It is necessary to clearly define the immune signatures predictive of clinical protection against BoHV-1 in cattle. Variation in responses observed among vaccines may be associated with the lack of specific parameters outlined for their production, inadequate quality control and/or use of inefficient adjuvants. Inappropriate handling and storage are also factors that negatively affect the immunogenicity of vaccines [21]. Here, we use registered vaccines that have been commercially available for many years. Two of these vaccines are manufactured abroad, while the other two are manufactured in Uruguay and exported to various countries. Presumably, the commercial vaccines have undergone rigorous quality controls; however, 3 out of the 4 vaccines evaluated here did not generate satisfactory levels of antibodies to ensure that they would protect against infection in the field with BoHV-1, or we must assume that the protective response is not based exclusively on the production of detectable levels of neutralizing antibodies.

The humoral responses of the vaccines differed from previous studies which detected higher levels of neutralization antibodies against BoHV-1 by SN test [20,21,22]. In particular, Anziliero et al. [20] found a level of protection against BoHV-1 between 80 and 100% in eight commercial vaccines tested in Brazil. In these studies, the differences found may have been due to the brands of vaccines tested, the control strain used (they used the Cooper strain and we used the Los Angeles strain), as well as the viral incubation neutralization time, which, in our case, was 24 h [19], while they used two hours. Another study showed that SN sensitivity varied greatly depending on the challenge virus used in the test, particularly when results against each virus were considered individually [15]. In addition, there is a difference between the vaccine strains (we have to say which strain the vaccines use) and the Los Angeles samples used in the SN test, which can also influence the detected antibody titer. Thus, the comparison of results with previous studies has to be done with reservations.

Recently, other researchers have analyzed the serological response against BoHV-1 by comparing live vaccines (MLV) and inactivated vaccines [23,24]. The best protective response was observed in heifers vaccinated with live modified thermosensitive virus [24]. However, in a meta-analysis where the prevention of abortion in cattle following vaccination against BoHV-1 was reviewed, it was found that both MLV and inactivated vaccines decrease abortion risk. Regardless of potential risk, the overall effect of MLV vaccines was a protective effect against abortion [9]. In Uruguay, live vaccines are prohibited and therefore their use in our experiment was not possible.

Additionally, we tested the immunogenicity of vaccine D applied simultaneously with a Clostridial vaccine (Group 5), as well as when applied with an FMD vaccine (Group 6). The practice of applying two or more vaccines simultaneously is widely used in Uruguay and in various countries of the region to optimize the handling of animals. However, there is a lack of field experiments that demonstrate that this practice does not affect the quality of the expected immune response for each immunogen used. Here, it did not generate acceptable levels of protection against BoHV-1 (Table 2), though a significant difference was observed in the IgG titers at day 30 in both groups (5 and 6) compared to the control. This shows that the simultaneous immunization could have affected the initial activation of the immune system (primary response), but then this difference disappeared at day 60. This allows us to assume then that there would be no interference with the response against BoHV-1, at least with the titers of generated antibodies. Despite this, the quality and duration of the humoral and cellular immune response to vaccination should be studied in future experiments.

It is not uncommon for farmers to use a different brand of vaccine for the second dose of the vaccine. Anecdotally, this practice is associated with a higher-quality immune response than applying the same brand in both immunizations. Here, in group 7, we applied vaccine A in the first dose and vaccine D in the second dose. The levels of protection determined by the neutralizing antibody titers for vaccine A and D were 86% and 20%, respectively, while the combination of brands A + D (group 7) achieved protective titers in 33% of animals (Table 1). Therefore, it was found that the response of vaccine A was affected by changing the brand, while the response of vaccine D was favored in that group.

Most immune responses associated with vaccination are controlled by specific T cells of a CD4+ helper phenotype, which mediate the generation of effector antibodies, cytotoxic T lymphocytes (CTLs), or the activation of innate immune effector cells [25]. Although T helper cells play a central role in the induction of a protective immune response against infections from viral pathogens, Th2 cells producing interleukin (IL)-4, IL-13 and IL-5 can be detrimental in an infection. Several murine studies firmly established that IL-4 regulates B cells for the secretion of IgG1 antibodies, whereas interferon-γ stimulates the expression of IgG2a antibodies, rendering either isotype an indicator of the underlying Th2 or Th1 response in mice [26]. It has also been shown in cattle that IgG1 expression is positively regulated by IL-4, and IgG2 expression is positively regulated by IFN-gamma [17]. In examining the results of the in-house ELISA employed here, high levels of IgG2 were detected in sera of animals vaccinated with all four vaccines and combinations of vaccines, except vaccine B, where no antibody levels were detected. This significant increase in IgG2 would suggest that a Th1 response profile was activated after vaccination. In addition, we had previously found an IgG2 response in animals naturally infected with BoHV-1 [27], as well as animals vaccinated against BoHV-1; the immune response profile is Th1 with a predominance of IgG2.

In conclusion, our study showed that the four commercial BoHV-1 multivalent vaccines tested produced different antibody levels in cattle after the complete vaccination protocol. Only one vaccine fulfilled the minimum requirements of immunogenicity, conferring adequate neutralizing antibodies in the vaccinated animals. When we evaluate the IgG response, high IgG2 titers were found for all vaccines that had a detectable serological response, indicating a Th1 response profile that is adequate for antiviral response. Taken together, these results demonstrate that the level of neutralizing antibodies alone may not be an adequate parameter for determining protection against BoHV-1. A virus challenge was not carried out in this study and would be an appropriate follow-up study to this work.

## Figures and Tables

**Figure 1 vaccines-09-00408-f001:**
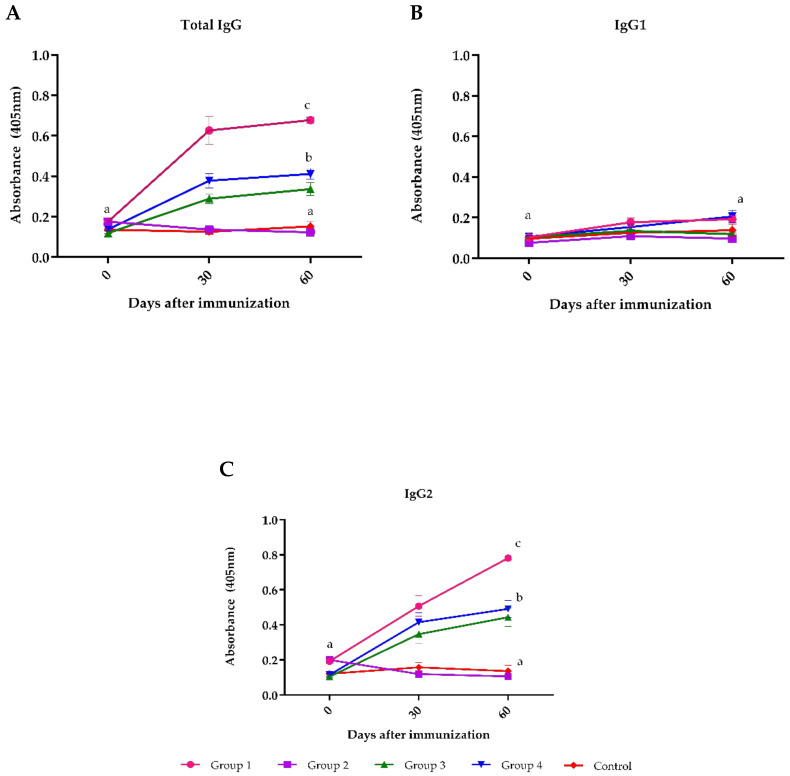
Antibodies to BoHV-1 evaluated by ELISA in sera of cattle immunized with four commercial BoHV-1 vaccines. Group 1 = vaccine A; group 2 = vaccine B; group 3 = vaccine C; group 4 = vaccine D; and Control group (not vaccinated). Mean titers ± standard errors (SEM) are depicted. Letters (a, b and c) highlight significant differences (*p* < 0.05) between groups. **A**: Total IgG antibodies. **B**: IgG1 subclass antibodies. **C**: IgG2 subclass antibodies.

**Figure 2 vaccines-09-00408-f002:**
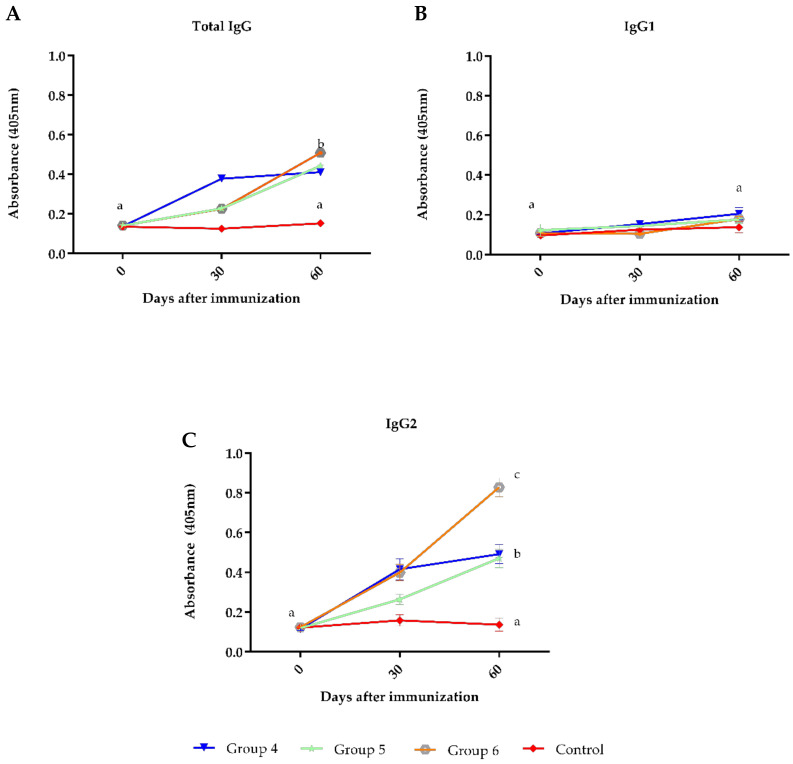
Antibodies to BoHV-1 evaluated by ELISA in sera of cattle immunized with commercial reproductive vaccines applied simultaneously with Clostridial or FMD vaccines. Group 4 = vaccine D; group 5 = vaccine D + Clostridial vaccine; group 6 = vaccine D + FMD vaccine and Control group (not vaccinated). Mean titers ± standard errors (SEM) are shown. Letters (a, b, c and d) highlight significant differences (*p* < 0.05) between groups. **A**: Total IgG antibodies. **B**: IgG1 subclass antibodies. **C**: IgG2 subclass antibodies.

**Figure 3 vaccines-09-00408-f003:**
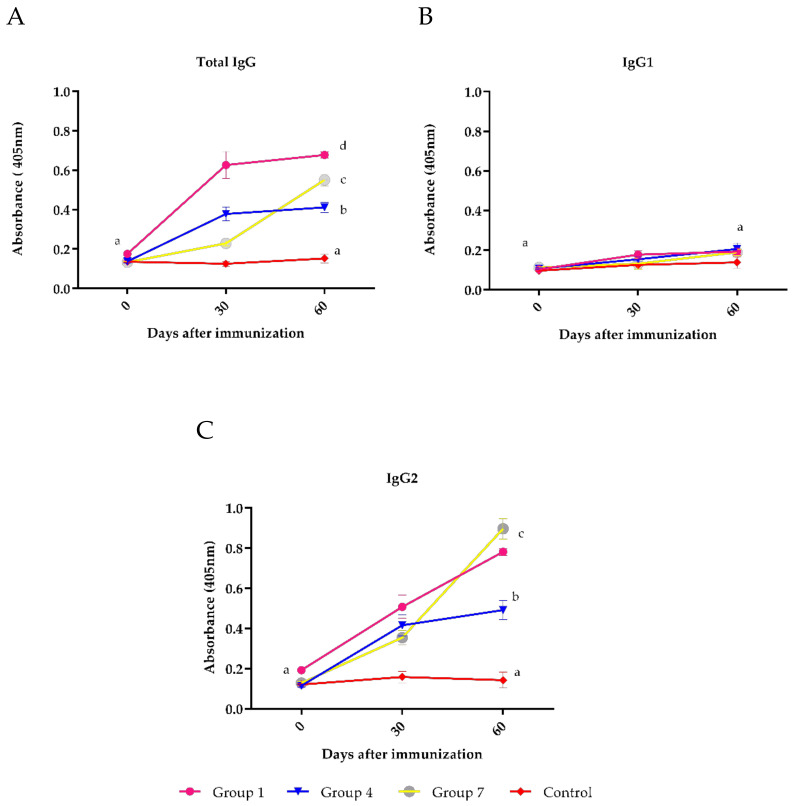
Antibody titers evaluated by ELISA in bovine sera immunized on day 0 with vaccine A and day 30 with vaccine D. Group 1 = vaccine A; group 4 = vaccine D; group 7 = vaccine A + D; and Control group = not vaccinated. Mean titers ± standard errors (SEM) are shown. Letters (a, b and c) highlight significant differences (*p* < 0.05) between groups. **A**: Total IgG antibodies. **B**: IgG1 subclass antibodies. **C**: IgG2 subclass antibodies.

**Table 1 vaccines-09-00408-t001:** Groups of animals (*n* = 15) and vaccines used in this study.

Groups	Vaccines
1	A
2	B
3	C
4	D
5	D + Clostridial vaccine
6	D + FMD vaccine
7	A (1st dose) + D (2nd dose)
Control	Non-vaccinated

FMD: Foot and Mouth Disease.

**Table 2 vaccines-09-00408-t002:** Neutralization antibodies for BoHV-1 at 30 and 60 days post-vaccination (dpv) in cattle immunized with four commercial vaccines.

Groups	Vaccines	Number of Animals	Reagents	
			30 dpv	60 dpv
1	A	15	4 * #4(8)	13 *^,a^ #2(8) #2(12) #2(16) #1(24) #5(32) #1(64)
2	B	15	0 *	0 ^b^
3	C	15	1 * #1(8)	1 ^b^ #1(8)
4	D	15	2 * #2(8)	3 ^b^ #2(8) #1(16)
5	D + Clostridial	15	3 * #2(8) 1(12)	3 ^b,c^ #2(16) #1(24)
6	D + FMD	15	1 * #1(8)	1 *^,b^ #1(8)
7	A (1a dose) + D (2a dose)	15	2 * #2(8)	5 *^,c^ #1(8) #1(12) 3(16)
Control	Non-vaccinated	15	0 *	0 *^,b^

* Number of reactive animals; # Number of animals and neutralizing antibody titers detected (in parentheses). All animals were seronegative for SN test at day 0. The different letters (^a^,^b^ and ^c^) show significant differences (*p* < 0.05).

## Data Availability

Data available on request from the authors.

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
