# Peer review of "Field Evaluation of Commercial Vaccines against Infectious Bovine Rhinotracheitis (Ibr) Virus Using Different Immunization Protocols"

_vaccines, 2021, doi:10.3390/vaccines9040408_

Round 1

Reviewer 1 Report

The authors present an interesting study that has compared the capacity of four commercially available vaccines for bovine herpesvirus 1 (BoHV-1) to induced protection in cattle. They also examined the influence of co-administering other vaccines with a selected BoHV-1 vaccine. While another component of the study, looked at the impact of using two of the BoHV-1 vaccines in combination, as this is reported to be a common practice in the Uruguayan cattle industry. Overall the study should be of interest to those working on this important bovine virus. The conclusions drawn are generally supported by the data. I have made some suggestions below for the authors to consider.

Line 19 suggest revision “ubiquitous in cattle populations and is associated with several”

Line 22 suggest revision “The objective of this study was to evaluate the performance of four commercially available BoHV-1 vaccines in cattle, especially those marketed in central and south America.”

Line 82: How were the animals included in the trial selected? What was the serological status of the herd? Had any of the vaccines been used previously in the herd management?

Line 106 Is the name of the adjuvant included in Vaccine D known? If so, please included this in the description.

Line 114 I would strongly suggest the authors add the names of the commercial vaccines to Table 1. While I can understand why they have elected not to specifically associate the commercial names to the formulations used in the treatment groups, in listing the individual components of each formulation it seems it should be relatively easy for someone to “match” these up. More importantly, it is essential that sufficient methodology be provided to enable someone to replicate the experiment describes here. Lack of knowledge of what formulations were used prevents this. In making the decision to publish the data in the manuscript the authors are obligated to meet this key criterion.

Line 154 – What statistical approach was used to compare the groups in Table 2. Presumably, the variable use in this analysis was the number or proportion of animals with an SNT ≥8?

In Table 2 – the authors only report the data for animals with SNT titers ≥8. I assume they have done so as in the discussion, it is mentioned that an SNT titer ≥8 is considered protective (Line 211).

This raises a couple of questions:

  1. Were all cattle SNT negative at Day 0?
  2. What proportion of cattle had SNT titers above 0 at Day 60?

Question 2 is of particular interest as Figure 1, suggests that the total antibody responses were very consistent within groups. While accepting that directly comparing SNT to ELISA is problematic, it is equally problematic not to report SNT titers less than 8. For example, if all of the animals for which SNT data is not report had SNT titers of 4, this would be a very different scenario if in comparison they were 0. Perhaps the authors could consider providing the complete SNT dataset as a supplemental file.

I would also suggest describing the use of the SNT ≥8 as the cut-off for protection in the relevant methods section. This ensures the reader understands why it is used as the cut-off for data in Table 1.

Line 158 suggest revision “demonstrated”

Line 159-160 suggest revision: “in comparison to the”

Line 166 Figure 1 – suggest a revision of y-axis title to “Absorbance (405nm)”

It is also not clear which symbols represent each treatment group.

What do the data points and error bars represent? Mean with SEM or SD?

Presumably, the data present in panels A to C are crude values, as the absorbance values at Day 0 are well above zero. Did the authors consider using the Day 0 values to adjust the values on Day 30 and Day 60?

I would make similar comments to those above for Figures 2 & 3 as well. In each figure, while the group number is provided and the treatment – it is not possible to distinguish the data points as the symbols are not included.

Line 216 Suggest the authors consider replacing “recognised” with “registered”

Line 219-220 While the vaccines may not have induced neutralising antibodies in the high proportion of animals, immunisation may still have a protective effect, albeit suboptimal.

Line 223 I would suggest the authors be very clear here what they mean when they refer to higher antibody levels. Does this mean SNT titers ≥8 or is this based on ELISA results?

Line 227 Would available genetic information for the LA and Cooper strains support this statement?

Line 237-250 – I certainly understand what the authors were trying to achieve in the co-administration of additional vaccines, particularly as this a common practice in industry. However, one point I think the authors should mention in this context is that none of the BoHV-1 vaccines tested in this study were monovalent. They all contain multiple targets. However, only the responses to the BoHV-1 component were measured in this study. Similarly, the responses of the targets of the two “competing” vaccines in Group 5 and Group 6 were not assessed. While BoHV-1 was the focus of the current study, it would be interesting to determine if there were responses to the clostridial and FMDV in these groups. Of course, without groups immunised with these vaccines alone, it would not be possible to determine these effects. However, if responses to these vaccines were extremely poor, it might suggest the immune system was overwhelmed by the repertoire of antigens across the two vaccines. Alternatively, strong responses in Group 6 to FMDV might indicate this vaccine was able to “out-compete” the BoHV-1 vaccine.

Line 252 I would suggest the authors replace “empirically” with “anecdotally”, as I believe this is based on non-scientific reports from industry.

This reported effect, is arguable similar to the now well-established effect referred to as prime/boost. However, for this to work the vaccines needs to be delivered via two different routes, eg intramuscular, then intranasal etc. However, as only inactivated vaccines were available for comparison it would seem this is not a viable strategy in Uruguay.

Lines 260-276 – Is the mouse isotype IgG2a the same as the bovine IgG2 isotype? Typically studies will refer to the IgG1/IgG2 ratio rather than looking at each in isolation. It is also somewhat unusual to see a Th1 type response from inactivated vaccines. However, it would not be unexpected from the study referred to with naturally infected animals, Ref 17. One might expect to see a biased Th2/antibody response to an inactivated vaccine, while or Th1/CMI response to an infection of MLV.

Reviewer 2 Report

Leites et al. investigated the capability of different commercially available vaccines to induce neutralizing, IgG1, and IgG2 antibodies in bovines. The study is well performed and well written and the results are of significance, especially considering that most of the tested vaccines could not induce neutralizing antibodies, which is worrisome. I have only a few suggestions.

- An important aspect to highlight in the introduction is that there are different subtypes of BoHV-1. Also, is there anything known about cross-reactivity between antibodies and viruses from different subtypes?

- Section 2.2. Did animals in groups 5 and 6 receive only one dose of each vaccine? Please, specify.

- Figures 1, 2, and 3. Since this journal allow the publication of color images for no extra cost, I strongly recommend to use colors to improve graph readability.

- Line 226. Cooper and Los Angeles belong to the same subtype and are highly identical and I don’t think that the antigenic differences between these two strains could explain such a big difference. Are the subtypes of the strains used for the vaccines known? If viruses used in the vaccines belong to a different subtype than the virus used for the neutralization assay and ELISA, the reduced immune response could be due to a partial cross-reactivity. This may have not been observed in other studies if there was a better antigenic match between the vaccine and the virus used for lab assays.

Minor:

- Line 19. Virus species name should be italicized.

- Lines 22-23. This sentence is a bit odd, please check the grammar.

- Line 39. I suggest using the official species name “Bovine alfaherpesvirus 1” (written in italics and without “type)

- Line 61. “Clostrial” sounds a bit weird in this sentence. Maybe you mean clostridial antigens?

- Line 63. Please, define more clearly what you mean with “reproductive vaccines”.

- Line 63. Odd sentence. Suggestion: “would interfere with the immune response against BoHV-1”.

- Line 68. “It has been showed”.

- Lines 109-110. Bacterial specie names have to be italicized.

- Section 2.4. Please, specify the subtype for the Los Angeles strain.

- Line 141. Typo: “then added”.

- Line 189. Typo: “changed”

- Lines 201-3. This sentence is a bit odd, please check the grammar.

- Line 207. Typo: “recognized”

- Line 208. This sentence is a bit odd, please check the grammar.

- Line 274. Typo: “animals naturally infected”

- Line 275. This sentence is a bit odd, please check the grammar.

Reviewer 3 Report

The manuscript presents valuable vaccine information on the bovine herpesvirus1.  The pathogen is of agricultural concern, worldwide but especially in Uruguay.  The study is very well done with valuable data.  The manuscript is overall well written but does have some errors that should be addressed. The study is complex, but the data well presented. It is disturbing that the B vaccine did not induce an antibody response while C and D resulted in a very poor overall response. The combination responses with the clostridial and FMD vaccines were examined with D which was apparently alone a poor vaccine. It is of concern that only one of these vaccines performed well.  It is of interest that the use of a second vaccine could result in a decent immune response.

There are several grammatical problems in the composition of the manuscript which should be addressed. Some of these are listed below.

Abstract

Line 22. . .was to evaluate. . .

Line 23 …of cattle vaccines of importance marketed in Central and South American

Line 31. . .induced significant, detectable levels of. . .

Introduction

Line 49. . .as undetectable introduction into herds ensuring. . .

Line 52. . .the Americas, Australia,

Line 57. . . in cattle, such as . . .

Line 61. . .immunogens, such. . .

Line 159. . . antibodies (p,0.05) relative to

Line 172. . .a significantly lower antibody response than. . .

Line 202. . .is not only of economic importance but also is a major. . .

  1. . . At the international level. . .
  2. . .while the other two. . .
  3. . .efficacies of the vaccines differed from previous studies. .
  4. . . for farmers to use a. . .
  5. . . Although T helper cells. . .against infections from viral pathogens, Th2 cells producing. . .

274 . . .also found IgG2 responses  in natural infections with BoHV-1. . .
